# Absolute IOP/EOP Estimation Models without Initial Information of Various Smart City Sensors

**DOI:** 10.3390/s23020742

**Published:** 2023-01-09

**Authors:** Namhoon Kim, Sangho Baek, Gihong Kim

**Affiliations:** 1Department of Civil Engineering and Environmental Sciences, Korea Military Academy, 574 Hwarang-ro, Nowon-gu, Seoul 01805, Republic of Korea; 2Department of Civil Engineering, Gangneung-Wonju National University, 7 Jukheon-gil, Gangneung-si 25457, Republic of Korea

**Keywords:** smart city sensor, camera sensor, position and orientation estimation, direct linear transformation, perspective projection model

## Abstract

In smart cities, a large amount of optical camera equipment is deployed and used. Closed-circuit television (CCTV), unmanned aerial vehicles (UAVs), and smartphones are some examples of such equipment. However, additional information about these devices, such as 3D position, orientation information, and principal distance, is not provided. To solve this problem, the structured mobile mapping system point cloud was used in this study to investigate methods of estimating the principal point, position, and orientation of optical sensors without initial given values. The principal distance was calculated using two direct linear transformation (DLT) models and a perspective projection model. Methods for estimating position and orientation were discussed, and their stability was tested using real-world sensors. When the perspective projection model was used, the camera position and orientation were best estimated. The original DLT model had a significant error in the orientation estimation. The correlation between the DLT model parameters was thought to have influenced the estimation result. When the perspective projection model was used, the position and orientation errors were 0.80 m and 2.55°, respectively. However, when using a fixed-wing UAV, the estimated result was not properly produced owing to ground control point placement problems.

## 1. Introduction

Cities have recently been transformed into smart cities to increase their survivability and improve the quality of life of their residents. The goals of smart cities are achieved through the use of various sensors to collect and analyze data [1]. Cameras are examples of optical sensors that are used in smart cities. Optical sensors are used to perform real-time actions such as traffic control, social safety, and disaster response [2,3]. For example, closed-circuit television (CCTV) cameras are already installed in many cities and play an important role. In Korea, the number of CCTV cameras installed is gradually increasing for purposes such as crime prevention and disaster monitoring.

However, the precision of the three-dimensional position of optical sensors has received little attention. In the case of CCTV, the position information is roughly provided. However, only latitude and longitude can be checked, and the orientation information is unavailable [4]. Furthermore, cameras used in smart cities are not standardized. Hence, people cannot easily use specific information about cameras. Determining the specifications and locations of these numerous sensors takes time, incurs administrative costs, and is impossible without the cooperation of various organizations. In addition, public officials in charge have a low perception of the significance of camera information. In summary, interior orientation parameters (IOPs) and exterior orientation parameters (EOPs), which are the most important information regarding a sensor, are difficult to use.

To extract EOP information from cameras, various photogrammetric methods can be used. Based on the collinearity equation, single-photo resection (SPR) is the most representative EOP estimation algorithm. In SPR, a solution is obtained by repeatedly adjusting three or more control points. SPR has been studied to increase its efficiency [5,6,7]. However, because SPR is sensitive to the initial values of EOPs, it is difficult to use if those values are not specified [8,9]. A quaternion-based SPR algorithm was proposed to solve the initial-value problem and the gimbal lock phenomenon [8]. However, only ground control points (GCPs) that meet certain criteria can be used [10], which poses a limitation. To estimate the camera’s EOPs, SPR algorithms based on the law of cosines [11] and the Procrustes algorithm [12,13,14] were proposed. The PnP algorithm, proposed by Fischler and Bolles, is a method of estimating the position and orientation of a camera using a point corresponding to a 3D object and a 2D image [15]. It estimates the EOPs of a camera using a perspective projection model and is used in a variety of fields including indoor positioning and robotics [16,17,18]. Nonetheless, these methods can be used while the camera’s interior orientation information is known. The objective of this study is to develop a method for estimating the camera’s IOPs (especially focal length)/EOPs in situations where the vendor has not performed camera calibration and new calibration is difficult to perform.

The direct linear transformation (DLT) model can be used to estimate the camera’s IOPs and EOPs simultaneously. The DLT model was compared to the collinearity model and the perspective model by Seedahmed and Schenk [19]. DLT parameters are used to express EOP parameters. The least-square solution (LESS) is commonly used to estimate DLT parameters [20]. This method is widely used in photogrammetry and computer vision because of its significant advantage in estimating the camera’s IOPs/EOPs using a simple formula [21,22]. However, the accuracy of the estimated IOPs/EOPs is inferior to that of physical models such as the collinearity equation or the coplanarity equation [23].

The EOP estimation method using the perspective projection model also produces good results. For example, a perspective projection model was used to estimate radial distortion values, principal distances, and EOPs [24,25,26,27]. The solution was discovered using the Gröbner basis or the Sylvester matrix.

Several algorithms have been developed to estimate camera EOPs and to calibrate cameras. Table 1 summarizes the features of each algorithm as well as whether the initial value is required. Algorithms that can produce results without prior camera information can be useful for extracting location information and camera parameter information from many optical sensors in urban areas. These algorithms have good performance. However, there have been few studies on the reliability of the results and the comparison of the performance of each algorithm.

Furthermore, some studies performed positioning estimation of various sensors using deep learning [28,29,30,31], but it is difficult to accept that it is close to the true value from a surveying standpoint.

Although many studies have been conducted in this way, previous studies have focused on the theoretical part rather than the application of actual data. Also, as far as we know, no research has been conducted that analyzes each algorithm using the same data. This study performed absolute position and orientation estimation as well as camera calibration for cameras with no initial information. We proposed and compared standardized algorithms that can be applied to a variety of camera sensors such as smartphones, drones, and CCTV without initial information. This study focuses on estimating IOP (principal distance)/EOP information. The main objectives were:Investigation and comparative analysis of IOP/EOP estimation models;Stability and accuracy analysis of IOP/EOP estimation models;Analysis of estimation results using practical optical sensor data.

## 2. Methodology

### 2.1. Camera Geometric Model

#### 2.1.1. DLT Model

A DLT model connects points in 3D space and 2D image planes using parameters. Because of their simplicity and low computational cost, DLT models are widely used in close-range photogrammetry, computer vision, and robotics. The mathematical model of planar object space and image space using homogeneous coordinates is given as
(1)$$\left[\begin{array}{c} x\\ y\\ 1 \end{array}\right]=\left[\begin{array}{cccc} L_{1} & L_{2} & L_{3} & L_{4}\\ L_{5} & L_{6} & L_{7} & L_{8}\\ L_{9} & L_{10} & L_{11} & L_{12} \end{array}\right]\left[\begin{array}{c} X\\ Y\\ Z\\ 1 \end{array}\right]$$
where *x* and *y* are image coordinates; *X*, *Y*, and *Z* are the object space coordinates; and *L_n_* is the DLT parameter. This model can be written as follows:(2)$$x=\frac{L_{1}X+L_{2}Y+L_{3}Z+L_{4}}{L_{9}X+L_{10}Y+L_{11}Z+L_{12}}+e_{x},\;y=\frac{L_{5}X+L_{6}Y+L_{7}Z+L_{8}}{L_{9}X+L_{10}Y+L_{11}Z+L_{12}}+e_{y}$$

At least six well-distributed GCPs are required to calculate the DLT parameters in Equation (2). The LESS can be used to determine the best DLT parameters. Lens distortion parameters can also be applied to the DLT model by using Equation (3) [32]:(3)$$x=\frac{L_{1}X+L_{2}Y+L_{3}Z+L_{4}}{L_{9}X+L_{10}Y+L_{11}Z+L_{12}}+dist_{x}+e_{x},\;y=\frac{L_{5}X+L_{6}Y+L_{7}Z+L_{8}}{L_{9}X+L_{10}Y+L_{11}Z+L_{12}}+dist_{y}+e_{y}.$$

The DLT parameter can be obtained using a variety of constraints. The most common DLT condition is $L_{12}=1$ (ordinary DLT, ODLT). Furthermore, the norm criterion ($L_{1}^{2}+L_{2}^{2}+\cdots +L_{12}^{2}=1$, norm criterion DLT, NDLT) was also presented [33].

#### 2.1.2. Perspective Projection Model

Equation (4) is a perspective projection camera model expressed by a homogeneous vector:(4)$$x\lambda x_{I}=PX_{W}$$
where $X_{W}$ is the 4 × 1 world point homogeneous vector (${\left[\begin{array}{cccc} X & Y & Z & 1 \end{array}\right]}^{T}$), $x_{I}$ is the 3 × 1 image point homogeneous vector (${\left[\begin{array}{ccc} x & y & 1 \end{array}\right]}^{T}$), and $P=\mathrm{diag}\left(f,f,1\right)\left[I\;|\;0\right]$ is the 3 × 4 homogeneous camera projection matrix. Principal point offset, pixel ratio, and skew can be applied using Equation (5):(5)$$\left[\begin{array}{c} fX+Zp_{x}\\ fY+Zp_{y}\\ Z \end{array}\right]=\left[\begin{array}{cccc} \alpha f & 0 & p_{x} & 0\\ 0 & \beta f & p_{y} & 0\\ 0 & 0 & 1 & 0 \end{array}\right]\left[\begin{array}{c} X\\ Y\\ Z\\ 1 \end{array}\right]$$
where ${\left[\begin{array}{cc} p_{x} & p_{y} \end{array}\right]}^{T}$ represents the coordinates of the principal point; α,β are the pixel ratios; and *s* is the skew parameter [20].

Figure 1 shows the process of converting two different coordinate systems (world coordinate system and camera coordinate system) using rotation and translation. The geometric camera model with camera rotation and translation is applied as follows [20]:(6)$$\lambda x_{I}=K\left[R\;|\;t\right]X_{W}=PX_{W}$$

### 2.2. Absolute Position/Orientation Estimation and Calibration Using Camera Models

#### 2.2.1. DLT Model

The camera position and parameters can be calculated using the DLT model [19]. Equations (7) and (8) show the DLT and perspective projection models, respectively.
(7)$$\lambda x_{I}=\left[\begin{array}{cccc} L_{1} & L_{2} & L_{3} & L_{4}\\ L_{5} & L_{6} & L_{7} & L_{8}\\ L_{9} & L_{10} & L_{11} & L_{12} \end{array}\right]X_{W}$$
(8)$$\lambda x_{I}=K\left[R\;|\;t\right]X_{W}=KR\left[I\;|-X_{O}\right]X_{W}$$
where $x_{I}$ is the homogeneous image coordinates vector, K is the calibration matrix, R is the rotation matrix, $X_{O}$ is the camera position vector, $X_{W}$ is the homogeneous object point vector, and I is the identity matrix.

From Equations (7) and (8), Equation (9) can be derived:(9)$$\left[\begin{array}{cccc} L_{1} & L_{2} & L_{3} & L_{4}\\ L_{5} & L_{6} & L_{7} & L_{8}\\ L_{9} & L_{10} & L_{11} & L_{12} \end{array}\right]=KR\left[I\;|-X_{O}\right]$$

Equation (9) can be rewritten as follows:(10)$$KR=D,\;D=\left[\begin{array}{ccc} L_{1} & L_{2} & L_{3}\\ L_{5} & L_{6} & L_{7}\\ L_{9} & L_{10} & L_{11} \end{array}\right]$$

On the basis of Equations (9) and (10), the camera position and rotation matrix can be computed as follows:(11)$$X_{O}=-D^{-1}d,\;R=K^{-1}D$$

$X_{O}$ matrix means the camera position, and R matrix means the camera orientation. The camera calibration matrix *K* can be calculated by Equation (10) and Choleski factorization:(12)$$\left(KR\right){\left(KR\right)}^{T}=DD^{T}$$
(13)$$KK^{T}=DD^{T}$$

#### 2.2.2. Perspective Projection Model

Equation (14) is the geometric model of the perspective projection:(14)$$\lambda x_{I,i}=PX_{W,i}$$
where $x_{I,i}$ is the *i*th image point coordinates and $X_{W,i}$ is the *i*th world point coordinates. Let us assume a typical camera (skew parameter ≈ 0 and pixel ratio ≈ 1). The camera calibration matrix *K* will be $K=\mathrm{diag}\left(1,1,w\right)$ for $w=\frac{1}{f}$. With the assumptions, Equation (14) can be written as follows:(15)$$\lambda x_{I,i}=\left[\begin{array}{cccc} r_{11} & r_{12} & r_{13} & t_{x}\\ r_{21} & r_{22} & r_{23} & t_{y}\\ wr_{31} & wr_{32} & wr_{33} & wt_{z} \end{array}\right]X_{W,i}$$

Undistorted and distorted image coordinates are expressed by Equation (16), according to Fitzgibbon’s radial distortion model [34].
(16)$$p_{u}\approx \frac{p_{d}}{1+k_{1}r_{d}^{2}+k_{2}r_{d}^{4}+k_{3}r_{d}^{6}}$$
where *k* is the radial distortion parameter, $p_{u}={\left[\begin{array}{ccc} x_{u} & y_{u} & 1 \end{array}\right]}^{T}$ is an undistorted image point, $p_{d}={\left[\begin{array}{ccc} x_{d} & y_{d} & 1 \end{array}\right]}^{T}$ is a distorted image point, and $r_{d}^{2}=x_{d}^{2}+y_{d}^{2}$ is the radius of $p_{d}$ for distortion center. The image point can be written as follows:(17)$$x_{I,i}={\left[{\begin{array}{ccc} x_{i} & y_{i} & 1+k_{1}\left(x_{i}^{2}+y_{i}^{2}\right)+k_{2}{\left(x_{i}^{2}+y_{i}^{2}\right)}^{2}+k_{3}\left(x_{i}^{2}+y_{i}^{2}\right) \end{array}}^{3}\right]}^{T}$$

In this step, we will use the properties of the skew-symmetric matrix. The skew-symmetric matrix ${\left[a\right]}_{\times }$ of vector $a={\left[\begin{array}{ccc} a_{1} & a_{2} & a_{3} \end{array}\right]}^{T}$ is defined as:(18)$${\left[a\right]}_{\times }=\left[\begin{array}{ccc} 0 & -a_{3} & a_{2}\\ a_{3} & 0 & -a_{1}\\ -a_{2} & a_{1} & 0 \end{array}\right]$$

By the property of the skew-symmetric matrix, ${\left[x_{i}\right]}_{\times }x_{i}=0$ can be obtained as
(19)$$\left[\begin{array}{ccc} 0 & -\left(1+k_{1}r_{i}^{2}+k_{2}r_{i}^{4}+k_{3}r_{i}^{6}\right) & y_{i}\\ 1+k_{1}r_{i}^{2}+k_{2}r_{i}^{4}+k_{3}r_{i}^{6} & 0 & -x_{i}\\ -y_{i} & x_{i} & 0 \end{array}\right]\left[\begin{array}{cccc} p_{11} & p_{12} & p_{13} & p_{14}\\ p_{21} & p_{22} & p_{23} & p_{24}\\ p_{31} & p_{32} & p_{33} & p_{34} \end{array}\right]\left[\begin{array}{c} X_{i}\\ Y_{i}\\ Z_{i}\\ 1 \end{array}\right]=0$$

The third row of Equation (19) can be rewritten as follows:(20)$$-y_{i}\left(p_{11}X_{i}+p_{12}Y_{i}+p_{13}Z_{i}+p_{14}\right)+x_{i}\left(p_{21}X_{i}+p_{22}Y_{i}+p_{23}Z_{i}+p_{24}\right)=0$$

The seven parameters, $p_{11},\;p_{12},\;\ldots ,\;p_{24}$, are unknown. If seven GCPs are obtained, the seven equations can be expressed in the matrix form as
(21)$$\begin{array}{c} M_{1}v_{1}=\left[\begin{array}{cccccccc} -y_{1}X_{1} & -y_{1}Y_{1} & -y_{1}Z_{1} & -y_{1} & x_{1}X_{1} & x_{1}Y_{1} & x_{1}Z_{1} & x_{1}\\ -y_{2}X_{2} & -y_{2}Y_{2} & -y_{2}Z_{2} & -y_{2} & x_{2}X_{2} & x_{2}Y_{2} & x_{2}Z_{2} & x_{2}\\ \vdots & \vdots & \vdots & \vdots & \vdots & \vdots & \vdots & \vdots \\ \vdots & \vdots & \vdots & \vdots & \vdots & \vdots & \vdots & \vdots \\ -y_{7}X_{7} & -y_{7}Y_{7} & -y_{7}Z_{7} & -y_{7} & x_{7}X_{7} & x_{7}Y_{7} & x_{7}Z_{7} & x_{7} \end{array}\right]\left[\begin{array}{c} p_{11}\\ p_{12}\\ p_{13}\\ p_{14}\\ p_{21}\\ p_{22}\\ p_{23}\\ p_{24} \end{array}\right]=0 \end{array}$$

By decomposing Matrix
M1 through SVD, the last column of matrix V can be selected as a solution. A constant value λ is needed to obtain the actual solution because the norm of the chosen solution is fixed at 1. Equation (22) shows the solution vector $v_{1}^{\prime }$ from the last column vector of matrix V.
(22)$$v_{1}^{\prime }={\left[\begin{array}{cccccccc} v_{1} & v_{2} & v_{3} & v_{4} & v_{5} & v_{6} & v_{7} & v_{8} \end{array}\right]}^{T}=\frac{1}{\lambda }{\left[\begin{array}{cccccccc} p_{11} & p_{12} & p_{13} & p_{14} & p_{21} & p_{22} & p_{23} & p_{24} \end{array}\right]}^{T}.$$

As $p_{11},p_{12},p_{13}$ are elements of the rotation matrix, Equations (23)–(25) can be established:(23)$$p_{11}^{2}+p_{12}^{2}+p_{13}^{2}=1,$$
(24)$$p_{11}^{2}+p_{12}^{2}+p_{13}^{2}=\lambda^{2}v_{1}^{2}+\lambda^{2}v_{2}^{2}+\lambda^{2}v_{3}^{2}=1,\;\mathrm{and}$$
(25)$$\lambda =\sqrt{\frac{1}{v_{1}^{2}+v_{2}^{2}+v_{3}^{2}}}.$$

Using Equations (22) and (25), elements corresponding to 2/3 of the *P* matrix $\left(p_{11},p_{12},p_{13},p_{14},p_{21},p_{22},p_{23},p_{24}\right)$ can be obtained.

Let a 3 × 3 submatrix of matrix P be $P^{\prime }$. Matrix $P^{\prime }$ can be written as Equation (26):(26)$$P^{\prime }=\left[\begin{array}{ccc} p_{11} & p_{12} & p_{13}\\ p_{21} & p_{22} & p_{23}\\ p_{31} & p_{32} & p_{33} \end{array}\right]=KR$$

Herein, R is the rotation matrix of the camera. The three rows of matrix $P^{\prime }$ are perpendicular because matrix K is a diagonal constraint matrix. In addition, the norm of the first and second-row vectors of matrix $P^{\prime }$ are the same; thus, Equations (27)–(30) established:(27)$$p_{11}p_{21}+p_{12}p_{22}+p_{13}p_{23}=0,$$
(28)$$p_{31}p_{11}+p_{32}p_{12}+p_{33}p_{13}=0,$$
(29)$$p_{31}p_{21}+p_{32p_{22}}+p_{33}p_{23}=0,\;\mathrm{and}$$
(30)$$p_{11}^{2}+p_{12}^{2}+p_{13}^{2}-p_{21}^{2}-p_{22}^{2}-p_{23}^{2}=0.$$

Let $p_{31}=\delta$. Subsequently, $p_{32}$ and $p_{33}$ can be parameterized for δ by using Equations (27) and (28). The results are given as follows:(31)$$p_{32}=-\frac{\delta \left(p_{11}p_{23}-p_{21}p_{13}\right)}{p_{12}p_{23}-p_{22}p_{13}}=-\delta c_{1},$$
(32)$$p_{33}=-\frac{\delta \left(p_{21}p_{12}-p_{11}p_{22}\right)}{p_{12}p_{23}-p_{22}p_{13}}=-\delta c_{2}$$
where $c_{1}=\frac{p_{11}p_{23}-p_{21}p_{13}}{p_{12}p_{23}-p_{22}p_{13}},\;c_{2}=\frac{p_{21}p_{12}-p_{11}p_{22}}{p_{12}p_{23}-p_{22}p_{13}}$. The remaining unknown parameters are $p_{31},\;p_{34},\;k_{1},\;k_{2},\;k_{3}$. These five unknown parameters can be obtained using the second row of Equation (20):(33)$$\left(1+k_{1}r_{i}^{2}+k_{2}r_{i}^{2}+k_{3}r_{i}^{2}\right)\left(p_{11}X_{i}+p_{12}Y_{i}+p_{13}Z_{i}+p_{14}\right)-x_{i}\left(p_{31}X_{i}+p_{32}Y_{i}+p_{33}Z_{i}+p_{34}\right),$$
(34)$$M_{2}v_{2}=v_{observation},$$
where
(35)$$M_{2}=\left[\begin{array}{ccc} x_{i}X_{i}+c_{1}x_{i}Y_{i}+c_{2}x_{i}Z_{i} & x_{i} & -r_{i}^{2}\left(p_{11}X_{i}+p_{12}Y_{i}+p_{13}Z_{i}+p_{14}\right) \end{array}\right.\;\;{\left.\begin{array}{cc} -r_{i}^{4}\left(p_{11}X_{i}+p_{12}Y_{i}+p_{13}Z_{i}+p_{14}\right) & -r_{i}^{6}\left(p_{11}X_{i}+p_{12}Y_{i}+p_{13}Z_{i}+p_{14}\right) \end{array}\right]}_{i\times 5}$$
(36)$$v_{2}={\left[\begin{array}{ccccc} \delta & p_{34} & k_{1} & k_{2} & k_{3} \end{array}\right]}_{\;i\times 5}^{\;T}.$$

Here, $M_{2}$ and $v_{2}$ have dimensions of 7 × 5 and 7 × 1, respectively, because seven GCPs are used. As a result, $v_{2}$ can be calculated using LESS as Equation (37):(37)$${\hat{v}}_{2}={\left(M_{2}^{T}PM_{2}\right)}^{-1}M_{2}^{T}Pv_{observation}.$$

The values of each element of matrix *P* and camera distortion parameters can be obtained using the equations described above. The principal distance is the final unknown parameter. The relationship between the first and last rows of matrix *P* can be used to calculate the principal distance. Based on Equation (14), Equations (38) and (39) are obtained by multiplying the first row of *P* by *w*:(38)$$w^{2}p_{11}^{2}+w^{2}p_{12}^{2}+w^{2}p_{13}^{2}-p_{31}^{2}-p_{32}^{2}-p_{33}^{2},$$
(39)$$w^{2}=\frac{p_{31}^{2}+p_{32}^{2}+p_{33}^{2}}{p_{11}^{2}+p_{12}^{2}+p_{13}^{2}}.$$

Finally, we can get focal length from Equation (39):(40)$$f=\frac{1}{w}=\sqrt{\frac{p_{11}^{2}+p_{12}^{2}+p_{13}^{2}}{p_{31}^{2}+p_{32}^{2}+p_{33}^{2}}}.$$

### 2.3. Equipment and Dataset

CCTV, unmanned aerial vehicles (UAVs), and smartphones, which can be used in smart cities, were used as target sensor platforms. Images were captured in a variety of environments with each sensor, and the estimation results were compared. EOPs were estimated and compared to a total station surveying result (position parameters, X, Y, Z) and the SPR result (orientation parameters ω, ϕ, κ). The process using the DLT models and the perspective projection model is illustrated in Figure 2. Figure 3 shows the camera platforms used in the experiments. Each sensor platform was calibrated using a checkerboard and a camera geometric model. 

Figure 4 shows images from each sensor platform. CCTV images were obtained from locations under conditions similar to those found in a smart city. Two drones were used to capture images: one in an oblique direction (rotary-wing UAV) and one in a nadir direction (fixed-wing UAV). The image acquisition conditions were investigated by comparing the results obtained from the two images. Smartphone images were acquired without any specific photographic conditions. The GCP positions on the images are denoted by yellow X marks.

Each image dataset has unique position and orientation properties. Although the height of the platform is clearly different, Figure 4a,b shows they have a similar orientation parameter of looking down diagonally. Figure 4b,c shows the camera mounted on a UAV, and while the Z value of the position is similar, the orientation parameter is noticeably different. Figure 4b examines the diagonal direction which can have a wide variety of GCPs, whereas Figure 4c shows the cause of the GCPs to be distributed on an almost constant plane. The Z diversity of GCP is particularly low in the park, which is the study’s target area. Finally, the smartphone image is captured by the user while holding the phone and looking to the side, which can differ significantly from the image orientation parameters of Figure 4a–c.

MMS + UAV hybrid point cloud data were used in this study to acquire the 3D location of GCPs and checkpoints (CKPs). The smart city point cloud was used because GCPs could be easily obtained without direct surveys. In this study, the georeferencing point cloud generated in Mohammad’s study [35] was used (as shown in Figure 5).

## 3. Experimental Results

### 3.1. Simulation Experiments

Before conducting an experiment using a real sensor, simulation experiments were performed to compare the performance of each algorithm using a 10 × 10 × 10 virtual grid. Thirteen virtual grid points were chosen as GCPs at random from a pool of 1000. The camera parameters, position, and rotation were calculated using 100,000 GCP combinations from a possible set of $C\left(1000,13\right)\approx 1.4849\times 10^{29}$. The camera parameters were set close to the actual camera parameters. Camera IOPs/EOPs and the coordinates of virtual points were also set based on the actual TM coordinate system. The set camera parameters and IOPs/EOPs values are listed in Table 2. The estimated values were directly compared with the true values. The reprojection error was calculated using 987 virtual points. Figure 6a depicts the virtual grid and the camera position. Figure 6b depicts the virtual grid, and Figure 6c depicts the virtual image generated by the virtual grid. The simulation environment was Win 11, Matlab R2022b.

Figure 7 shows a comparison of each-algorithm-estimated principal points and EOPs. Based on the Median value, all three algorithms were able to estimate the principal distance with an accuracy of 0.1 pixels or less. However, when comparing the maximum values, NDLT resulted in the least error. Figure 7c–e shows the camera orientation and position estimation results. It was shown that the perspective projection model, NDLT, and ODLT model showed good performance in order. All three algorithms showed an error of less than 1 degree. Figure 7f–h shows the camera position estimation results. The NDLT model and perspective projection model showed good performance, and the ODLT model also showed satisfactory performance. The maximum error when using the NDLT and the perspective projection model did not exceed 1 m, but when the ODLT was used, the maximum error was relatively large.

Figure 8 shows box plots of the mean reprojection error of each algorithm. When comparing the mean reprojection error, ODLT showed outstanding performance. The maximum error of the ODLT model did not exceed 0.5 pixels. NDLT and Perspective projection models also showed good performance, but the maximum errors were 1.71 pixels and 3.72 pixels, respectively.

It is interesting to note that the X, Y, and Z distributions of GCP also affect the quality of the estimation results. Aside from the distribution of GCPs on the image plane, the even distribution of GCPs in a 3D object space is critical [36,37]. GCPs were randomly selected on one plane as shown in Figure 9a, and GCPs were randomly selected on multiple planes as shown in Figure 9b, and the results were compared.

When GCPs were selected on only one plane, IOP/EOP estimation was not performed properly. Figure 10 is a visual representation of the results. Figure 10 shows the size and direction of the reprojection error, and it can be seen that a visually unacceptable error has occurred. None of the three algorithms produced significant estimation results. In addition to the reprojection error, the estimation results of the camera IOPs and EOPs were also unacceptable. As with many camera models, it is clear that the 3D distribution of GCPs is critical.

Table 3 shows the camera EOPs and the mean reprojection errors. It is confirmed that the camera parameter estimation and the reprojection results have remarkably improved. The X, Y, and Z errors of all three models were all less than 50 cm. In particular, in the case of perspective projection, it was confirmed that the size of positional error was more than twice as small as that of other models. Orientation error and mean reprojection error were also the smallest in the perspective projection model.

The degree of the 3D distribution can be determined by the distance between the camera and the object. Let us compare close-range photogrammetry with an object–sensor distance of about 20 m with aerial photogrammetry with a flight altitude of 200 m or more. Even GCP distributions with the same depth range can be treated as near-planar distributions in aerial photogrammetry [38]. As a result, GCPs must be carefully chosen by the sensor platform.

### 3.2. Practical Experiments

This section describes experiments in which the ODLT, NDLT, and perspective projection models were used to estimate the actual sensor position/orientation and principal distance. Sensor calibration was performed prior to the experiments to determine the IOP values of each sensor. However, IOPs can change for a variety of reasons. For example, the principal point varies due to lens group perturbation and may vary due to aperture and focus changes [39,40,41]. The value of the radial distortion parameter changes with the principal distance, making generalized modeling difficult [40]. The estimated radial distortion parameter value can also vary with the distance from the control points [42,43]. The camera was set to manual mode to control various factors; however, the micromechanism that operated the lens group was not. Therefore, in this study, a direct comparative analysis was only used to estimate the camera EOPs. The focal length was shown to examine the trend of the estimation result, but the principal point location and camera distortion parameters were not shown.

To examine the accuracy of the estimated sensor position, a virtual reference station VRS GPS survey was performed. Further, as the true value of EOPs, the SPR result based on sensor measurement and camera calibration can be used as the initial value. The accuracy of the orientation estimation result can be indirectly checked using the mean reprojection error (MRE) and the comparison with the SPR result. In this paper, both orientation parameters estimated by SPR and reprojection results are presented. The locations of the GCPs are marked in Figure 4. Pixel coordinates and ground coordinates of GCPs were applied to $x_{i}$ and $X_{w}$ to estimate $L_{1}$ to $L_{12}$ in Equation (7). Based on $L_{1}$ to $L_{12}$, $X_{O}$ and K matrices were estimated to estimate camera position, orientation, and principal distance. In addition, pixel coordinates of GCPs were applied to $x_{i}$ and $y_{i}$, and ground coordinates of GCPs were applied to $X_{i}$, $Y_{i}$, and $Z_{i}$ of Equation (20) to estimate $p_{11}$ to $p_{34}$ for camera position, orientation, and principal distance.

#### 3.2.1. CCTV

The CCTV image was used to estimate the camera principal distance and EOPs. The reprojection error was calculated for each image using the estimated IOPs/EOPs and 10 CKPs. Table 4 shows the estimated principal distance, whereas Table 5 shows the EOPs of the camera based on the CCTV image. The estimated camera position error for each method is shown in Figure 11a, and the rotation angle error is shown in Figure 11b. The MRE for each model is depicted in Figure 11c.

In the case of the position estimation error, the perspective projection model produced the most accurate estimation results. The position errors for the ODLT and NDLT models were 1.9856 m and 1.3951 m, respectively. The perspective projection had an error of 0.6336 m, allowing for a more accurate position estimation. In the case of the rotation angle estimation, the perspective projection model produced the best results, whereas ODLT produced a large error in the rotation angle. However, the reprojection results were consistent across all three models.

#### 3.2.2. UAV

Table 6 shows the calibrated and estimated principal distance of the UAV camera sensor. The estimated EOP errors and the reprojection errors are shown in Table 7. Interestingly, as a result of experimenting with images taken in the direction of nadir (fixed-wing UAV), an unacceptably large error occurred in the IOP and EOP estimation. It was estimated very differently from the principal distance calibration result, and rotation and position errors largely occurred in the case of EOP as well. In contrast, the experiment using the image taken in the oblique direction to understand the rotary-wing UAV showed acceptable results. This is related to the distribution of the GCPs described in Section 3.1. The image was taken at a high altitude (>200 m), but the height distribution of the GCPs was within 4.09 m. Because all GCPs and CKPs were on nearly the same plane, the reprojection results were not large, but proper IOP/EOP estimation was not performed.

Next, the results obtained using the rotary-wing UAV image are shown in Figure 12. Figure 12a,b depicts the camera position and the camera rotation angle errors, respectively. Figure 12c also displays the MRE. In terms of the camera position error, the perspective-projection-model-based algorithm performed the best. The DLT-based algorithms also produced acceptable estimation results with errors of less than 1.6960 m and 2.1053 m, respectively. The position estimation accuracy of the perspective projection model was within 0.7966 m. The ODLT model had a maximum rotation angle estimation error of 7.15°, but the other two algorithms were generally capable of accurate rotation angle estimation. All three algorithms had the MRE of fewer than 5 pixels.

#### 3.2.3. Smartphone

The calibration and estimated principal distance of a smartphone camera are shown in Table 8. Table 9 displays the estimated EOP errors and the reprojection errors. Figure 13a shows a comparison of each-algorithm-estimated camera position error, and Figure 13b shows the estimated orientation angle error. The MRE is depicted in Figure 13c. All three models produced accurate camera position estimation results. The same pattern was observed in the results of orientation estimation. However, the ODLT position and orientation estimation performance suffered significantly.

Overall, the perspective projection model showed good results. This is because the correlation between parameters affected the quality when DLT models were used. The results of the three algorithms had lower reliability compared to the results of camera calibration or SPR, which are widely used. However, there was not much difference between the camera calibration result and the SPR result, and it is enough to be used as an initial parameter value. Therefore, it is possible to estimate the IOP/EOPs of the sensor precisely by fusion with the camera calibration and SPR.

## 4. Discussion

### 4.1. Simulation Experiments

When estimating IOPs (the principal distance and the principal point) in the two DLT models, A and C components were used for the x component of IOPs, and B and C components were used for the y component of IOPs. When estimating EOPs, all A, B, and C components were used. Figure 14 shows the relationship between the ODLT parameters calculated with the total least square.

Overall, the DLT parameters correlated with each other in the case of ODLT. The correlation between the parameters of the A, B, and C block components was high in the case of NDLT. A strong relationship between parameters can reduce estimation precision and increase error [44,45]. Because of high correlation between DLT parameters, errors in some parameters may be used to correct other parameters, causing errors to propagate to the accuracy of the IOP/EOP estimation [46]. In this regard, when IOPs/EOPs are estimated using ODLT and NDLT, the correlation between parameters influences the result, potentially lowering the estimation accuracy. In particular, it is expected that the estimation accuracy of ODLT, which shows the overall correlation, will be lower than that of NDLT.

### 4.2. Practical Experiments

Experimental results using CCTV, UAV, and smartphone, camera EOPs estimation results showed good results in regards to the point-based perspective projection model, NDLT model, and ODLT model. In the case of camera position estimation, ODLT and NDLT showed similar results, but the position estimation error was slightly larger when ODLT was used. In the case of the rotation angle estimation, NDLT and the perspective projection model showed significantly better results than ODLT. In the case of the reprojection error, the three models showed similar results. It is judged that this is because the high correlation between DLT model parameters affects the estimation result, as analyzed in Section 3.1. In addition, since DLT parameters are for the purpose of connecting 3D points and image points, the amount of reprojection error is smaller than that of the perspective projection model, but the quality of camera position and orientation estimation results are analyzed to be inferior.

In general, it was possible to estimate IOPs/EOPs using three models, but good results were not obtained using nadir images acquired from a UAV. The position estimation errors were over 100 m, and the rotation angle estimation was not able to estimate a reasonable result. The camera IOPs estimation result was also less reliable. This is because GCPs are almost on the same plane due to high altitude imaging. The reprojection results seem reasonable, but this is because the distribution of CKPs is also on the same plane as presented in Section 3.1. Neither DLT nor the perspective projection model can be used in this environment, but it is more appropriate to use the classic SPR.

### 4.3. Contribution and Limitations

A study was carried out in this paper to estimate the IOPs/EOPs of an optical sensor in the absence of an initial value. Sensor positioning was performed using three different algorithms, and the results were confirmed to be different. Experiments were carried out using both real data and simulation levels. CCTV, UAV, and smartphones were used, and it was discovered that applying the three algorithms was difficult if the diversity of GCP was not secured.

The limitations of this study are discussed as follows. The first limitation of this study is dependent on the quality of the point cloud from which GCP can be acquired. Many MMS devices currently acquire city point clouds, but the quality of the point clouds varies. When uncalibrated MMS equipment is used, the location accuracy of the point cloud is greatly reduced, which has a direct impact on the optical sensor’s orientation/position estimation result. The simplification of the camera distortion parameters is the study’s second limitation. The tangential distortion parameter was ignored in this study, and the radial distortion parameter was assumed to be small. This study did not include cases with large lens distortion parameters, such as fisheye lenses. As a result, future research must investigate how the location accuracy of the point cloud is propagated to the estimation results. Furthermore, when using a lens with a high distortion parameter, a position and orientation estimation process must be developed. However, in a situation where sufficient GCP can be secured, for example, in the case of an indoor space where a point cloud is acquired with terrestrial LiDAR, effective results can be produced for estimating the position and orientation of the sensor. This research team is conducting additional research related to point cloud registration using these characteristics and expects to obtain interesting results.

## 5. Conclusions

In this study, the IOPs/EOPs of various smart city sensors were estimated using ODLT, NDLT, and the perspective projection models. MMS + UAV hybrid point cloud data were used to collect GCPs and CKPs. We tested two different images for each platform. In this study, camera IOPs were not used as true values because calibration results could vary depending on the experimental conditions and fine optical adjustment of the instrument was not possible. Instead, the obtained calibration and estimation results were presented in tables to confirm the trend of IOPs.

In general, the estimated camera EOP results are ranked in descending order: the results of the perspective projection model, NDLT model, and ODLT model. In the case of the camera position estimation, ODLT and NDLT produce similar results, but ODLT produces slightly larger position estimation errors. The maximum error in estimating the sensor’s position using the perspective projection model was 0.7966 m, and the average error was 0.6331 m. The ODLT and NDLT models had average errors of 1.6992 m and 1.2047 m, respectively. In the case of the rotation angle estimation, NDLT and the perspective projection model significantly outperforms ODLT. The average orientation angle errors for the perspective projection model and the NDLT model were 0.88° and 0.76°, respectively, and 3.07° for the ODLT. The three models produce similar reprojection error results. The average reprojection error of each model was 3.67 pixel, 2.14 pixel, and 2.93 pixel, respectively. 

Herein, three models were used to estimate IOPs/EOPs. However, results obtained from UAV-acquired nadir images are poor. The position estimation results exceed 100 m, and the rotation angle estimation result is not reasonable. The estimation of camera IOPs is also less reliable. Because of high altitude imaging, GCPs can be regarded as being almost on the same plane. The reprojection results appear to be reasonable, but this is because the distribution of CKPs is also on the same plane. Table 10 is the error summary table for each sensor platform.

Through this study, it is possible to quickly estimate the camera information, position, and orientation of various optical sensors distributed in a smart city. Because it uses the geometric characteristics of a frame camera, it can be applied not only to the optical sensor but also to the infrared camera. In addition, there is an advantage in that absolute or relative coordinates of various sensor platforms can be calculated. In particular, the results of this study can be significantly applied to indoor and underground spaces where positioning systems such as global navigation satellite systems cannot be used. This research team plans to apply the findings of this study to the coarse registration of point clouds in indoor space in a future study.

## Figures and Tables

**Figure 1 sensors-23-00742-f001:**
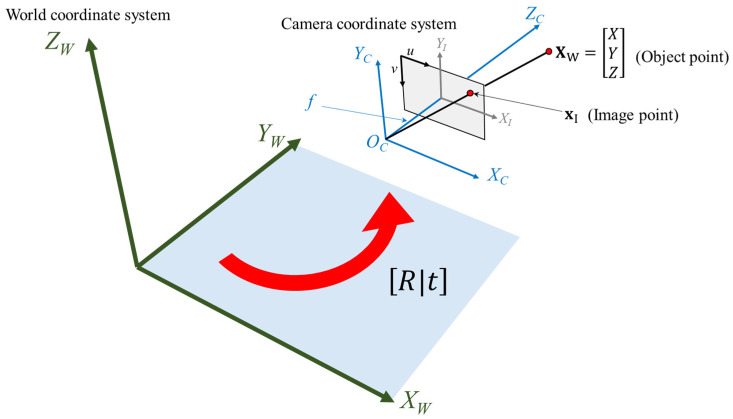
World coordinate system and perspective projection camera coordinate system.

**Figure 2 sensors-23-00742-f002:**
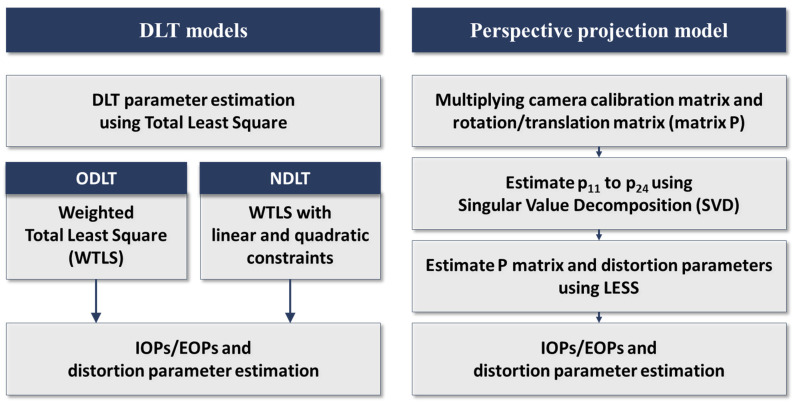
Process of estimating IOPs/EOPs using DLT model and perspective model.

**Figure 3 sensors-23-00742-f003:**
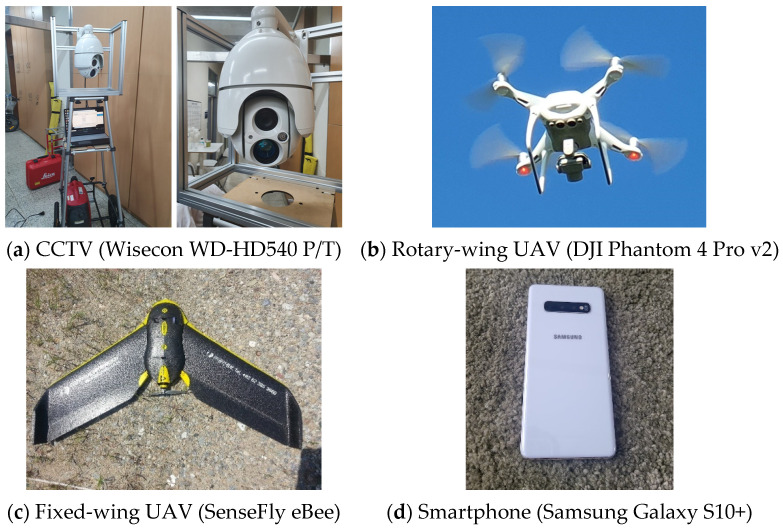
Camera sensor platforms.

**Figure 4 sensors-23-00742-f004:**
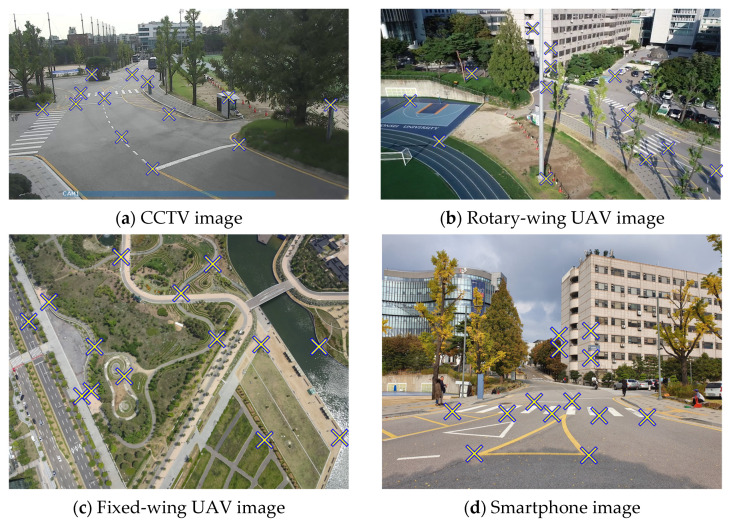
Images from each sensor platform.

**Figure 5 sensors-23-00742-f005:**
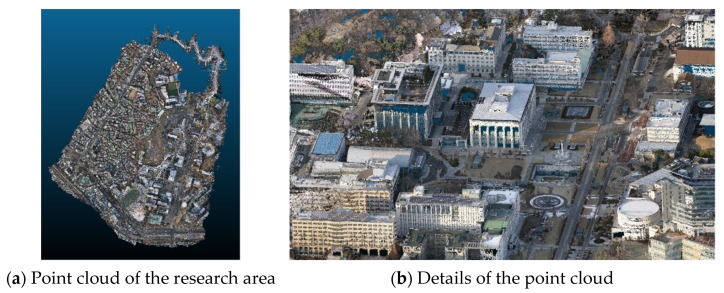
Hybrid point cloud for GCPs.

**Figure 6 sensors-23-00742-f006:**
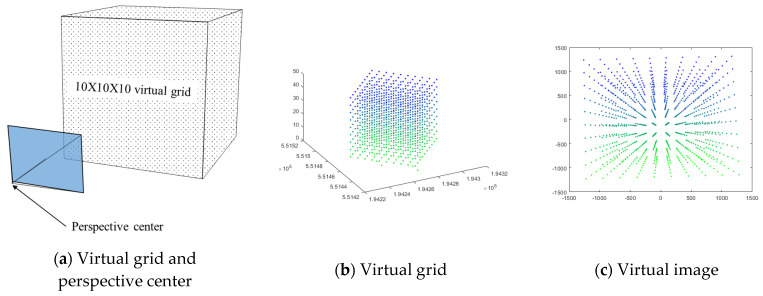
Virtual grid and virtual image.

**Figure 7 sensors-23-00742-f007:**
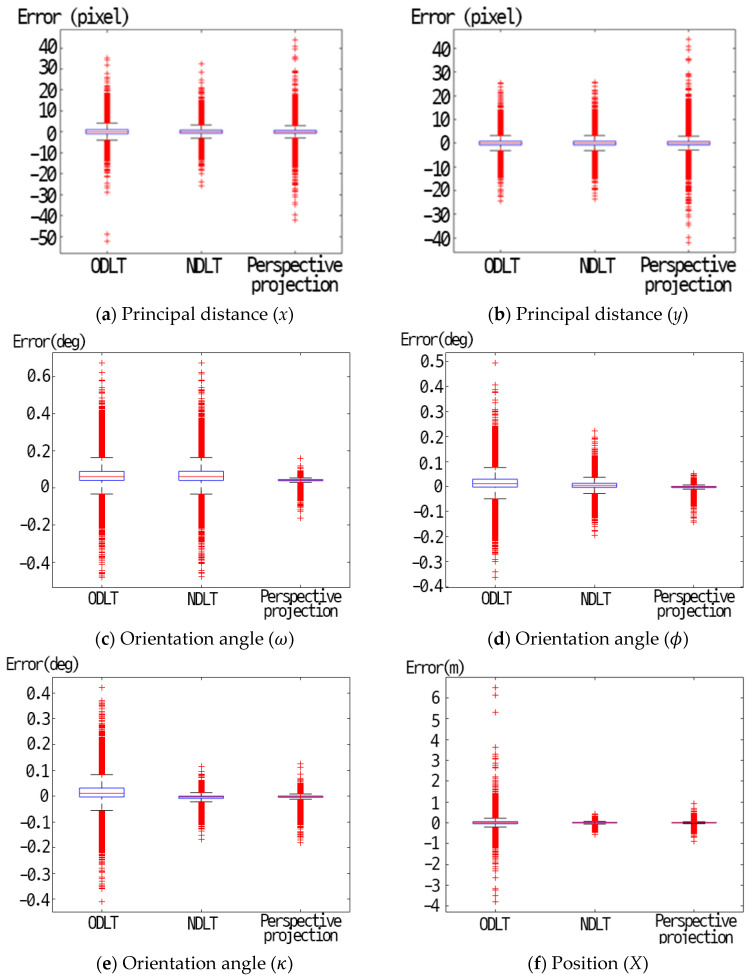
Accuracy assessment results using a 10 × 10 × 10 virtual grid.

**Figure 8 sensors-23-00742-f008:**
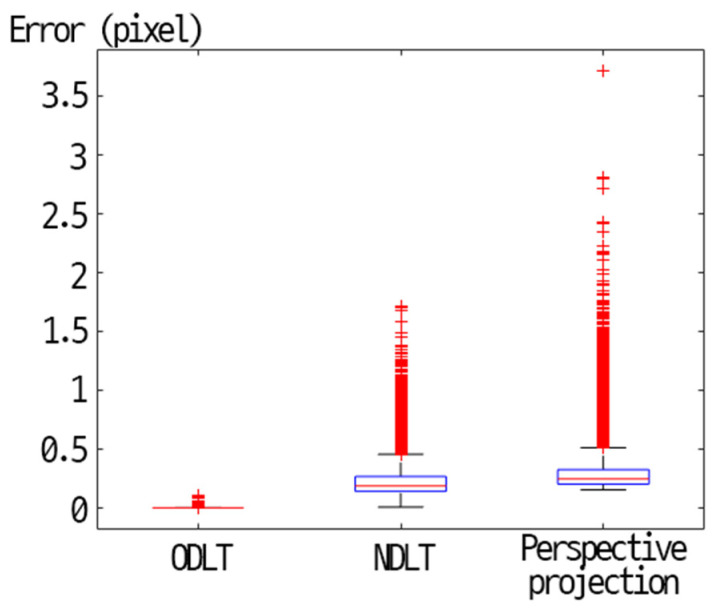
Box plots of mean reprojection error results.

**Figure 9 sensors-23-00742-f009:**
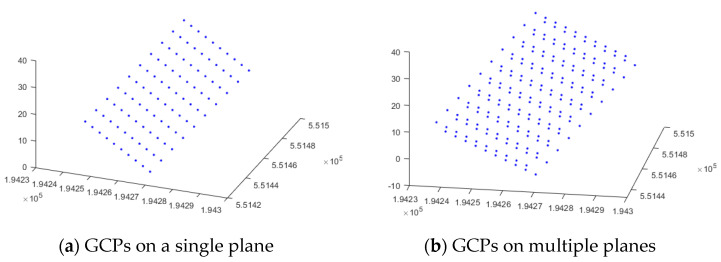
Three—dimensional distribution of GCPs.

**Figure 10 sensors-23-00742-f010:**
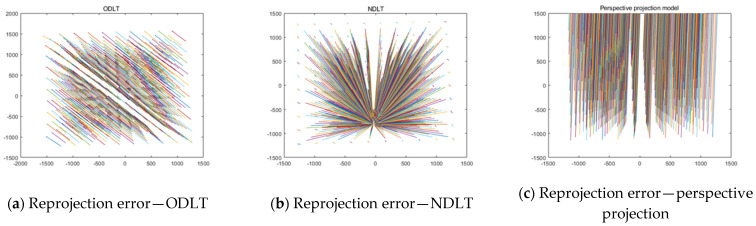
Reprojection error when using GCPs on one plane.

**Figure 11 sensors-23-00742-f011:**
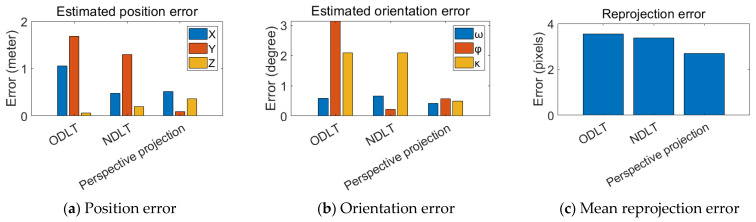
Estimated EOPs error (**a**,**b**) and mean reprojection error (**c**) for CCTV.

**Figure 12 sensors-23-00742-f012:**
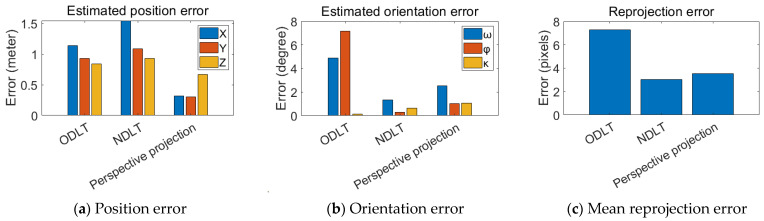
Estimated EOP error (**a**,**b**) and mean reprojection error (**c**) for UAVs.

**Figure 13 sensors-23-00742-f013:**
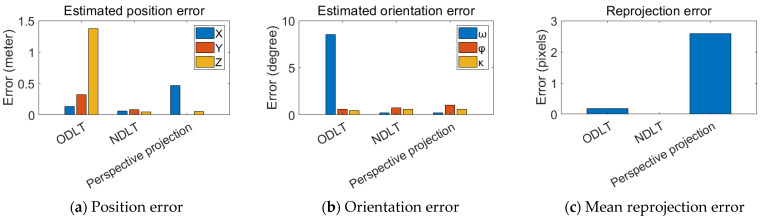
Estimated EOP error (**a**,**b**) and mean reprojection error (**c**) for a smartphone.

**Figure 14 sensors-23-00742-f014:**
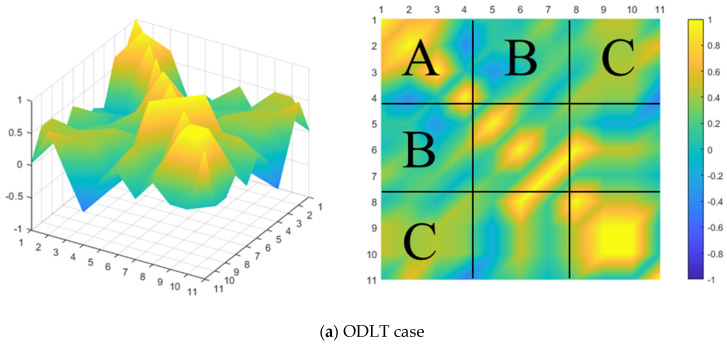
Visualizing the correlation of the DLT parameters.

**Table 1 sensors-23-00742-t001:** Characteristics of each camera position and orientation estimation algorithm.

Algorithms	Initial EOPs	Initial IOPs	Orientation Estimation
Ordinary SPR	Required and sensitive	Required	ω,φ,k
Quaternion SPR	Not required	Required	Rotation matrix
Law of cosine	Required but not sensitive	Required	Rotation matrix
Procrustes algorithm	Not required	Required	Rotation matrix
PnP algorithm	Not required	Required	Rotation matrix
DLT model	Not required	Not required	Indirectly estimate
Perspective projection model	Not required	Not required	Rotation matrix

**Table 2 sensors-23-00742-t002:** Camera parameters, IOPs, and EOPs.

Camera Parameters		Values
Principal distance		3500 pixels
Principal point	$x_{p}$	50 pixels
	$y_{p}$	20 pixels
Radial distortion parameter	$k_{1}$	−7.86 × 10^−9^
	$k_{2}$	6.92 × 10^−14^
	$k_{3}$	−1.29 × 10^−19^
Rotation angle	ω	94.045°
	ϕ	45°
	κ	1°
Camera position	$X_{O}$	194,200 m
	$X_{O}$	551,400 m
	$Z_{O}$	20 m

**Table 3 sensors-23-00742-t003:** Estimated camera EOP errors and mean reprojection errors.

Errors	X (m)	Y (m)	Z (m)	ω (°)	ϕ (°)	κ (°)	MRE (Pixels)
ODLT	0.48	0.21	0.35	0.1847	0.0896	0.1019	0.1847
NDLT	0.44	0.28	0.13	0.1858	0.0855	0.0858	0.1858
P.Prj.	0.21	0.11	0.00	0.0011	0.0561	0.0025	0.0011

**Table 4 sensors-23-00742-t004:** Calibration result and estimated principal distance of CCTV.

	$c_{x}$ (Pixels)	$c_{y}$ (Pixels)
Calibration result (Wisecon WD-HD540 P/T)	1627.14	1632.05
ODLT	1610.82	1650.28
NDLT	1648.60	1696.50
P.Prj.	1658.12	1658.12

**Table 5 sensors-23-00742-t005:** Estimated EOP and reprojection errors of CCTV.

Errors	X (m)	Y (m)	Z (m)	Position Error (m)	ω (°)	ϕ (°)	κ (°)	MRE (Pixels)
ODLT	1.0545	1.6814	0.0601	1.9856	0.58	3.14	2.09	3.55
NDLT	0.4767	1.2956	0.2011	1.3951	0.66	0.22	2.09	3.37
P.Prj.	0.5131	0.0965	0.3590	0.6336	0.41	0.57	0.49	2.69

**Table 6 sensors-23-00742-t006:** Calibration result and estimated principal distance of UAVs.

		$c_{x}$ (Pixels)	$c_{y}$ (Pixels)
Calibration result (DJI Phantom 4 Pro v2)		877.61	878.62
Rotary-wing image	ODLT	880.37	885.97
	NDLT	888.63	882.42
	P.Prj.	884.05	884.05
Calibration result (SenseFly eBee)		3648.49	3649.33
Fixed-wing image	ODLT	6051.15	6239.31
	NDLT	6315.11	6877.49
	P.Prj.	6871.42	6871.42

**Table 7 sensors-23-00742-t007:** Estimated EOP and reprojection errors of UAVs.

Errors		X (m)	Y (m)	Z (m)	Position Error (m)	ω (°)	ϕ (°)	κ (°)	MRE (Pixels)
Rotary-wing image	ODLT	1.1376	0.9332	0.8436	1.6960	4.89	7.15	0.14	7.27
	NDLT	1.5469	1.0843	0.9294	2.1053	1.35	0.30	0.64	3.05
	P.Prj.	0.3139	0.3031	0.6665	0.7966	2.55	1.03	1.06	3.53
Fixed-wing image	ODLT	5.1645	134.4646	91.2146	162.5654	74.45	40.83	21.61	5.42
	NDLT	7.0843	141.4162	100.1883	173.4544	28.43	75.87	107.07	3.31
	P.Prj.	47.8675	185.6578	164.1656	252.4093	37.26	18.08	7.71	5.79

**Table 8 sensors-23-00742-t008:** Calibration result and estimated principal distance of a smartphone.

	$c_{x}$ (Pixels)	$c_{y}$ (Pixels)
Calibration result (Galaxy S10+)	3219.51	3230.58
ODLT	3286.63	3240.36
NDLT	3272.71	3276.77
P.Prj.	3191.00	3191.00

**Table 9 sensors-23-00742-t009:** Estimated EOP and reprojection errors of a smartphone.

Errors	X (m)	Y (m)	Z (m)	Position Error (m)	ω (°)	ϕ (°)	κ (°)	MRE (Pixels)
ODLT	0.1318	0.3210	1.3729	1.4161	8.55	0.60	0.48	0.18
NDLT	0.0616	0.0814	0.0504	0.1138	0.22	0.74	0.61	0.00
P.Prj.	0.4662	0.0043	0.0528	0.4692	0.21	1.02	0.61	2.58

**Table 10 sensors-23-00742-t010:** Error summary for each sensor platform.

Errors		Position Error (m)	ω (°)	ϕ (°)	κ (°)	MRE (Pixels)
CCTV image	ODLT	1.9856	0.58	3.14	2.09	3.55
	NDLT	1.3951	0.66	0.22	2.09	3.37
	P.Prj.	0.6336	0.41	0.57	0.49	2.69
Rotary-wing image	ODLT	1.6960	4.89	7.15	0.14	7.27
	NDLT	2.1053	1.35	0.30	0.64	3.05
	P.Prj.	0.7966	2.55	1.03	1.06	3.53
Fixed-wing image	ODLT	162.5654	74.45	40.83	21.61	5.42
	NDLT	173.4544	28.43	75.87	107.07	3.31
	P.Prj.	252.4093	37.26	18.08	7.71	5.79
Smartphone	ODLT	1.4161	8.55	0.60	0.48	0.18
	NDLT	0.1138	0.22	0.74	0.61	0.00
	P.Prj.	0.4692	0.21	1.02	0.61	2.58

## Data Availability

Data not available due to the law of Korean government (ACT ON THE ESTABLISHMENT AND MANAGEMENT OF SPATIAL DATA).

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
