# Peer review of "Absolute IOP/EOP Estimation Models without Initial Information of Various Smart City Sensors"

_sensors, 2023, doi:10.3390/s23020742_

Round 1

Reviewer 1 Report

Absolute IOP/EOP Estimation Models without Initial Information of Various Smart City Sensors 

1. Particularly in the methodology section equations from 1 to 37 need detailed illustrations.

2.  In the manuscript flow of the research work should be addressed correctly. directly from the methodology section 3 experimental result and discussions arises. There is a need of one particular section in between to highlight the detailed steps in carrying out various research operations.

3. Section 3.1 and 3.2 need mode detailed explanations of the facts.

4. The current state of the conclusion is not clear so require reformations with discussions about clear findings rather than general discussions.

Author Response

The research team is extremely grateful to the reviewers who took the time to read the manuscript. Our manuscript was able to contain more rich content as a result of the comments that highlighted the key points. A response to comments is included in the attached document. We hope our responses are at the level of the reviewer.

Reviewer 2 Report

 In this work, some methods for estimation of interior orientation parameters and exterior orientation parameters in optical camera equipment for smart city applications are analyzed, where simulated and experimental results are presented. In general, the work is interesting, however, some issues have to be addressed to clarify and ensure contributions, applicability, and reproducibility.

In the introduction section, the contribution is unclear. Do you propose a new method or use an existing method? Which are the motivations to propose/use it? Which are the novelties/advantages?

In sections 2.1 and 2.2, there are many equations, some of which are referenced. Which are yours? Which are the mathematical contributions? How are they used in section 3?

For section 2.3, please indicate the impact of using other equipment and dataset.

Justify the values presented in lines 203-208. Which is the impact of using other values? Please provide other results.

The results of Figure 6 are not discussed. Are they good? Each subfigure in Figure 6 should be discussed in terms of its boxplot’s parameters. The methods used cannot be reproduced, please provide more details, including the computational cost and software/hardware used.

Include a general diagram/flowchart to describe the application of your proposed method.

Please complement in another way the results presented in Figure 10, e.g., by changing the color map and its scale.

For the results presented in section 3.2, indicate how your method/equations were used to obtain the results presented. Also, include intermediate results in order to observe how the method is going applied and its results at each step.

Please summarize in a table the main findings in each dataset/equipment.

In order to put in a better and wider context your work and contribution, please compare your proposal with other works (discussion section). Also, discuss the limitations of your work.

Please add quantitative findings to the conclusion section.

Increase the font size in the figures.

Update references.

Author Response

(The authors gave the same response as above.)

Round 2

Reviewer 1 Report

The current state of the manuscript may be considered for the publication.

Reviewer 2 Report

Comments and suggestions have been properly addressed. This reviewer recommends the manuscript acceptance.